# Dexamethasone and OLT1177 Cooperate in the Reduction of Melanoma Growth by Inhibiting STAT3 Functions

**DOI:** 10.3390/cells12020294

**Published:** 2023-01-12

**Authors:** Alberto Dinarello, Taylor S. Mills, Isak W. Tengesdal, Nicholas E. Powers, Tania Azam, Charles A. Dinarello

**Affiliations:** 1Department of Medicine, University of Colorado, Aurora, Denver, CO 80045, USA; 2Department of Internal Medicine, Radboud Institute of Molecular Life Sciences (RIMLS), Radboud University Medical Center, 6525 GA Nijmegen, The Netherlands

**Keywords:** melanoma, Inflammasome, STAT3, dexamethasone

## Abstract

The NLRP3 inflammasome is a multimolecular complex that processes inactive IL-1β and IL-18 into proinflammatory cytokines. OLT1177 is an orally active small compound that specifically inhibits NLRP3. Here, B16F10 melanoma were implanted in mice and treated with OLT1177 as well as combined with the glucocorticoid dexamethasone. At sacrifice, OLT1177 treated mice had significantly smaller tumors compared to tumor-bearing mice treated with vehicle. However, the combined treatment of OLT1177 plus dexamethasone revealed a greater suppression of tumor growth. This reduction was accompanied by a downregulation of nuclear and mitochondrial STAT3-dependent gene transcription and by a significant reduction of STAT3 Y705 and S727 phosphorylations in the tumors. In vitro, the human melanoma cell line 1205Lu, stimulated with IL-1α, exhibited significantly lower levels of STAT3 Y705 phosphorylation by the combination treatment, thus affecting the nuclear functions of STAT3. In the same cells, STAT3 serine 727 phosphorylation was also lower, affecting the mitochondrial functions of STAT3. In addition, metabolic analyses revealed a marked reduction of ATP production rate and glycolytic reserve in cells treated with the combination of OLT1177 plus dexamethasone. These findings demonstrate that the combination of OLT1177 and dexamethasone reduces tumor growth by targeting nuclear as well as mitochondrial functions of STAT3.

## 1. Introduction

The incidence of cutaneous melanoma is progressively rising [1,2]. The etiology of this type of cancer correlates with UV light exposure, ageing, environmental factors, and mutational load [3]. The mutations that lead to melanoma progression mainly affect the *BRAF* gene and its most common variant is the V600E substitution that activates the MAPK pathway, resulting in uncontrolled cell proliferation and inhibition of apoptosis [4,5,6]. Several in vitro melanoma cell lines are characterized by the *BRAF^V600E^* mutation, and, in the present analysis, we took advantage of the 1205Lu human line, which is characterized by this *BRAF* variant [7,8,9]. An important aspect in melanoma progression is the inflammatory status of the tumor. High and repeated exposures to UV light stimulate the recruitment of macrophages and neutrophils into the exposure area of the skin, with the subsequent upregulation of CCR2, ATF2, CCL8, MMP9 and other proteins involved in inflammation [10,11,12].

Considering the role of *BRAF* mutations in the pathogenesis of melanoma, treatments mostly target BRAF, MAPK, and ERK in order to dampen the deleterious effects of the constitutively active BRAF/MEK/ERK cascade [13]. However, more recent therapeutic strategies also target the immunosuppressive checkpoints that impair T cell-mediated anti-tumor mechanisms. Specific checkpoint inhibitors of CTLA-4 (ipilimuman) and PD-1 (pembroliozumab and nivolumab) are effective anti-tumor therapeutics [14,15,16]. Although the anti-tumor effects of checkpoint inhibitors have significantly prolonged life, they are often associated with life-threatening side-effects [17,18]. As reviewed by Palmieri and Carlino, these side effects consist in the activation of autoimmune diseases which include thyroid dysfunction, hypophysitis, hepatitis, colitis, pneumonitis, sarcoidosis, and arthritis [19]. For these reasons, an alternative strategy to suppress melanoma-induced inflammation is required.

Recent studies also reveal that tumor-specific NOD-like receptor protein 3 (NLRP3) is highly expressed and is active in melanoma [20,21]. Therefore, based on mouse studies, the NLRP3 inflammasome has become a promising target for cancer treatment, and successfully blocks the progression of several types of cancer [22,23,24,25]. Of these is the β-sulfonyl nitrile compound OLT1177, a NLPR3-specific inhibitor [26]. In humans, oral administration of OLT1177 is safe during a Phase I study [26]; oral OLT1177 reduced the pain and swelling in gouty arthritis [27], and increased the ejection fraction in patients with heart failure [28].

In mice, OLT1177 reduces melanoma growth and has an additive anti-tumor effect when combined with anti-PD-1 [21,29]. OLT1177 also reduces tumor levels of active IL-1α [21] and is highly expressed human melanoma [21,30] as well as in mouse B16F10 melanoma [21]. In addition, OLT1177 also prevents the conversion of the inactive IL-18 precursor to active IL-18. IL-18 has several biological properties and includes the production of IFNγ from T cells. IL-18-dependent induction of IFNγ reduces the effects of check-point inhibitors in the treatment of several types of cancer [31,32,33,34]. In addition to inducting IFNγ, IL-18 also promotes metastasis [35,36,37].

Dexamethasone is often used to reduce the side-effects of chemotherapy [38,39,40] and, in particular, the inflammation, as well as the ‘autoimmune’ nature of the checkpoint inhibitors [41,42]. Notably, the anti-inflammatory properties of dexamethasone enhance the antitumoral effects of several drugs [43,44] and have also direct antitumoral effects [45,46,47,48].

In the present work, we examined the anti-tumor properties of OLT1177 in mice implanted with B16F10 melanoma, as well as in in vitro melanoma cell cultures. We were particularly interested in the phosphorylations of tyrosine 705 and serine 727 of STAT3, which is a highly active transcription factor in tumors [49,50,51]. We also sought to assess the possible anti-tumor effects and mechanisms of dexamethasone itself and when combined with OLT1177 treatment. The mechanisms we studied included the in vitro metabolic effects of OLT1177, dexamethasone, and their combination, as well as STAT3-dependent transcription.

## 2. Materials and Methods

### 2.1. Cell Culture and Treatments

1205Lu and B16F10 melanoma cells were cultured, respectively, with RPMI and DMEM, supplemented with 10% FBS, 100 units/mL penicillin, 0.1 mg/mL streptomycin. Cells were maintained in a humidified 5% CO_2_ atmosphere at 37 °C. The cells were plated at 0.5 × 10^6^ per well in 6-well plates and allowed to adhere overnight. The following day, cells were treated with 10 μM dexamethasone, 10 μM OLT1177 or a combination of the two; in addition, the cells were cultured with or without 10 ng/mL recombinant IL-1α. Supernatants and samples for protein and RNA extraction were collected after 24 h.

### 2.2. Cytokine Measurements

Cytokine levels were measured by specific DuoSet ELISAs according to the manufacturer’s instructions (R&D Systems, Minneapolis, MN, USA).

### 2.3. In Vivo Model

Animal protocols were approved by the University of Colorado Animal Care and Use Committee. C57BL/6 wild type mice were purchased from The Jackson Laboratory (Bar Harbor, ME, USA). B16F10 cells (2 × 10^5^) were mixed with Matrigel (Corning) and implanted subcutaneously in the hind quarter of mice. Tumor size was recorded every day. After implantation, mice were fed either an OLT1177-enriched diet (7.5 mg OLT1177 per kg of food) or a standard food diet [21,25,52]. Dexamethasone (10 mg/kg) or PBS were injected intraperitoneally every three days. Mice were sacrificed 15 days after implantation and the tumors were removed for molecular analysis. Tumor volume was calculated using the formula *V* = (*LW*^2^)/2, where *L* is the length of the longest tumor dimension parallel to the skin containing the tumor midpoint, *W* is the length of the tumor dimension perpendicular to *L* and parallel to the skin, and *V* is the volume expressed in cubic millimeters (mm^3^). Dimensions were measured by electronic caliper on restrained mice. Tumor volumes were determined without knowledge of the experimental groups.

### 2.4. Extracellular Flux Analyzer (SeaHorse)

OCR and ECAR were measured by SeaHorse XFe96 Analyzer (Agilent Technologies, Santa Clara, CA, USA) using the SeaHorse XF96 Mito Stress Test and Glycolysis Stress Test, following the manufacturer’s instructions. The day before the test, either 1205Lu or B16F10 were added to 96-well Seahorse cell culture microtiter plates at a density of 40,000 cells per well suspended in SeaHorse XF Base Medium pH 7.4 with 1 mM HEPES (Agilent Technologies). The wells were pre-coated with Corning Cell-Tak (Life Sciences) according to the manufacturer’s instructions. The medium was supplemented with 2 mM L-glutamine for the Glycolysis Stress Test assay. The plate was incubated at 37 °C for 1 h in a non-CO_2_ incubator. OCR was assessed at baseline. After ECAR baseline measurements, for the Glycolysis Stress Test assay, glucose, oligomycin A, and 2-deoxy-glucose (2-DG) were added sequentially to each well to reach the final concentrations of 10 mM, 2 μM, and 50 mM, respectively. Glycolysis was calculated by subtracting the last ECAR measurement before glucose addition from the maximum ECAR measurement before oligomycin injection. The glycolytic capacity was calculated by subtracting the last ECAR measurement before glucose from the maximum ECAR measurement before 2-DG injection. OCR data are expressed as pmol of oxygen per minute. Oligomycin, FCCP, and rotenone were added sequentially to each well to reach the final concentrations of 1 μM, 2 μM, and 0.5 μM, respectively.

### 2.5. Protein Extraction and Western Blotting

1205Lu and B16F10 cells were cultured as previously described [21]. Primary tumors were collected from tumor-bearing mice. All cells were lysed in RIPA buffer (Sigma, Ronkonkoma, NY, USA) supplemented with protease and phosphatase inhibitors (Roche, Basel, Switzerland), centrifuged at 13,000× *g* for 30 min at 4 °C, and the supernatants were obtained. Protein concentration was determined in the clarified supernatants using Bio-Rad protein assay (Bio-Rad Laboratories, Santa Rosa, CA, USA). Electrophoresis was performed on Mini-Protean TGX 4–20% gradient gels (Bio-Rad Laboratories) and transferred to nitrocellulose 0.1 μm membranes (GE Water & Process Technologies, Trevose, PA, USA). Membranes were blocked in 5% rehydrated non-fat milk in TBS-Tween 0.5% for 1 h at room temperature. Primary antibodies for STAT3 (1:1000, 9132, Cell Signaling, Danvers, MA, USA), pSTAT3(Y705) (1:1000, 9131, Cell Signaling), pSTAT3(S727) (1:1000, 9134, Cell Signaling) and glucocorticoid receptor (1: 1000, 3660, Cell Signaling) were used in combination with peroxidase-conjugated secondary antibodies. A primary antibody against β-Actin (1:5000, sc-47778 HRP, Santa Cruz Biotechnology, Dallas, TX, USA) was used to assess protein loading.

### 2.6. mRNA Isolation and Quantitative Real Time Reverse Transcription PCR (RT-qPCR)

Total RNA was extracted from 1205Lu cell and from melanoma samples with TRIzol reagent. cDNA synthesis was performed using High-Capacity cDNA Reverse Transcription Kit (Applied Biosystems, Waltham, MA, USA) according to the manufacturer’s protocol. qPCRs were performed in triplicate with SYBR Green Master Mix (Applied Biosystems) by means of QuantvStudio^TM^ 3 Real-Time PCR System (Applied Biosystem). The amplification protocol was performed according to the manufacturer’s instructions: briefly, the protocol consists of 10 min at 95 °C for polymerase activation, followed by 40 cycles of 15 s at 95 °C (denaturation) and 60 s at 60 °C (annealing and extension). *18s* and *ACTB* were used as internal standards in each mouse and human sample, respectively. The sequences of the primers used are listed in Appendix A.

### 2.7. Scratch Assay

1205Lu cells were plated on a 24-well plate at a concentration of 0.3 × 10^6^ per well and were allowed to proliferate into a monolayer for 24 h. Prior to the scratch, one plate was pre-incubated for 2 h with human recombinant IL-1α. The monolayer was scratched with a sterile pipet tip (200 μL), washed with serum free RPMI, and photographed with an Olympus IX81 spinning disk microscope (0 h). Cells were then treated with 10 ng/mL IL-1α, 10 μM dexamethasone, 10 μM OLT1177 or 10 μM dexamethasone plus 10 μM OLT1177. After 24 h, the monolayers were photographed with the same microscope.

### 2.8. Statistical Analysis

Significance of differences was evaluated with Student’s *t*-test or ANOVA test using GraphPad Prism (GraphPad Software Inc., La Jolla, CA, USA). Statistical significance was set at *p* < 0.05.

## 3. Results

### 3.1. Melanoma Growth Is Significantly Reduced by Dexamethasone and OLT1177

To assess whether dexamethasone influences cancer growth and augments the anti-tumor properties of OLT1177, we subcutaneously implanted B16F10 melanoma cells in C57BL/6 mice and followed tumor growth for 15 days. One group of mice was fed standard food and PBS was injected intraperitoneally every 3 days (vehicle group); mice of another group received intraperitoneal 10 mg/kg of dexamethasone every three days. A third group was fed with OLT1177 containing food for the entire duration of the experiment (OLT group) as described in Tengesdal et al. [21]. The fourth group was fed OLT1177 enriched food and received intraperitoneal dexamethasone every three days. Tumor volumes were measured every day. The tumor growth curves show that vehicle tumors grew larger than tumors in either dexamethasone or OLT1177-treated mice. As shown in Figure 1, either OLT1177- or dexamethasone-treated mice exhibited three-fold smaller tumors compared to the vehicle (Figure 1A,B; Table 1) (*p* < 0.05). No significant differences were detected between OLT1177 and dexamethasone treatments. However, mice treated with OLT1177 plus dexamethasone showed the greatest reduction in tumor volume compared to the vehicle and to either treatment only (Figure 1A,B; Table 1). The reduction in tumor size observed in the combination group suggests that the two compounds inhibit tumor growth, and have an additive effect when administered together to tumor-bearing mice.

A major transcription factor involved in cancer progression is STAT3 [53,54,55]. STAT3 is transcriptionally activated upon the induction of the IL-1/IL-6/JAK-STAT3 pathway [29], which triggers the phosphorylation of Y705. Additionally, STAT3 can be phosphorylated at the level of S727, which mainly regulates mitochondrial functions of STAT3 [56,57]. Although STAT3 is also regulated by glucocorticoids, these hormones can positively or negatively affect the activity of STAT3 [58,59]. As shown in Figure 1D,E, phosphorylation of S727 and Y705 in the tumors are significantly reduced by the combination of OLT1177 and dexamethasone compared to mice treated with the vehicle, whereas total STAT3 is unaffected. Western blot analysis (Figure 1C–F) of tumors extracted from these mice confirm that the combination does not affect the overall levels of STAT3 protein. Thus, we conclude that the treatments, by dampening the levels of pSTAT3(Y705) and pSTAT3(S727), inhibit the activities of STAT3 in the tumor, which correlate with the reduction in tumor growth. As shown in Appendix A, the expression levels of the glucocorticoid receptor (GR), which binds dexamethasone, and of Fkbp5, a GR target gene, are not significantly affected by the treatments (Appendix A). To have a broader understanding of tumor phenotype upon dexamethasone and OLT1177 treatments, we decided to measure the expression levels of several important transcripts involved in tumor progression. Hif1a encodes for a highly active transcription factor in melanoma [60,61], and we could not see statistical differences in Hif1a expression levels among the treatments (Appendix A). Melanoma growth is also determined by the ability to generate blood vessels that feed the tumor [62]. For this reason, we measured the levels of expression of Vcam and Icam, two endothelial markers [63], and Vegf, which promotes endothelial growth and has a positive effect on melanoma progression [64]. All these transcripts are significantly downregulated in the dexamethasone + OLT1177 group when compared to the vehicle (Appendix A). Vcam is also significantly downregulated in the dexamethasone group compared to the vehicle, but not in the OLT1177 group compared to the vehicle (Appendix A), whereas Icam and Vegf are significantly downregulated in the dexamethasone + OLT1177 group compared to the dexamethasone only (Appendix A).

Next, we analyzed the functionality of STAT3 by measuring the expression levels of selected STAT3 target genes. *Socs3* encodes for an inhibitor of STAT3 that belongs to the negative feedback loop of JAK/STAT3 axis, and is the first transcript to be induced when STAT3 is active [65]. *Socs3* is upregulated in dexamethasone-treated tumors, but significantly downregulated in tumors from mice treated with the combination (Figure 2A). *Klf4* is also a target of STAT3 [66] and its expression is downregulated by OLT1177 (*p* = 0.0713) and in combination with dexamethasone (*p* < 0.05), either when compared to the vehicle or dexamethasone alone. These findings support the concept that combined treatment has a greater negative effect on STAT3-dependent transcriptional activity than OLT1177 (Figure 2B). We also examined STAT3-driven mitochondrial transcription by *Mt-nd1* and *Mt-nd4*. As shown in Figure 2C,D, STAT3-driven mitochondrial transcription is significantly downregulated by OLT1177 (*p* < 0.05) and, more significantly (*p* < 0.01), by the combination when compared to the vehicle. Dexamethasone alone does not affect the expression of these genes when compared to the vehicle. OLT1177 and dexamethasone + OLT1177 groups show a downregulation of *Mt-nd1* and *Mt-nd4* when compared to dexamethasone alone, suggesting that OLT1177 has a greater impact on the expression of these genes than dexamethasone alone.

### 3.2. Dexamethasone and OLT1177 Specifically Inhibit Two Phosphorylations of STAT3

To better dissect the molecular effects of OLT1177 and dexamethasone at a cellular level, we used in vitro cultures of 1205Lu human melanoma. Since inflammation in the tumor microenvironment includes a marked increase in IL-1α production [67], we examined the effects of dexamethasone and OLT1177 on 1205Lu cells either with or without 10 ng/mL of recombinant IL-1α. After 24 h, levels of IL-6 in the supernatants revealed a three-fold increase (Figure 3A, *p* < 0.001). As depicted in Figure 3B,C, respectively, there is a highly significant increase in the mRNA levels of *SOCS3* and pSTAT3 S727. Because the IL-1-dependent activation of STAT3 canonical activity has been reported [29], we additionally observed that pSTAT3(S727) level also increases in cells exposed to IL-1α (Figure 3D). First, we measured the levels of IL-1β in 1205Lu supernatants after 24 h of treatment. Either dexamethasone or OLT1177 significantly decreased the levels of IL-1β, only when 1205Lu cells were stimulated with IL-1α (Appendix A). Next, we performed a scratch test, since OLT1177, dexamethasone, and their combination reduced tumor growth. The scratch test is an in vitro assay often used to evaluate wound healing, cell proliferation, and cell migration [68]. The assay physically removes a path of confluent cells in the middle of the culture with a micropipette tip. The cell cultures are photographed at time 0 and 24 h after the scratch [68]. The ratio between scratched area at the beginning and at the end of the experiment gives a quantification of cell proliferation toward the scratched area. For example, higher values correspond to a high level of proliferation, whereas a low ratio value represents decreased proliferation. This assay has been used with human bone osteosarcoma epithelial cells [69], gastric carcinoma cell lines [70], melanoma [71], and the epidermoid carcinoma cell line [72]. As depicted in Figure 3E–H, in unstimulated 1205Lu cells there was no significant difference between the gap of control and treated cells. On the other hand, when 1205Lu cultures were incubated with IL-1α, we observed that dexamethasone and combination with OLT1177 significantly reduced cell proliferation and the scratch closure (Figure 3G,H).

As shown in Figure 4A, cells treated with IL-1α showed a reduction in pSTAT3 (S727) when treated with 10 μM dexamethasone, as well as a reduction in pSTAT3(Y705) when exposed to 10 μM OLT1177 (Figure 4B). As shown, both serine and tyrosine phosphorylations are significantly downregulated by the combined treatment (Figure 4A,B), whereas total STAT3 protein levels are not affected by the treatments (Figure 4C,D). Notably, we could not see any effects on tyrosine or serine phosphorylation on cells without IL-1α stimulation (Appendix A), and GR expression levels are not affected by the treatments (Appendix A). The OLT1177-dependent downregulation of STAT3 tyrosine phosphorylation is consistent with the data obtained by Tengesdal and collaborators [29]. However, we report in the present study that dexamethasone reduces serine phosphorylation. The regulation of STAT3 phosphorylation by OLT1177 and dexamethasone showed similar outcomes in murine melanoma and in human melanoma cell lines (Figure 3), implicating an evolutionary conservation of the STAT3 phosphorylation mechanisms between mouse and human. To properly access the STAT3 activities, we measured the expression levels of selected STAT3 target genes. Direct STAT3 nuclear transcriptional activity was measured with *SOCS3* expression, which is significantly downregulated (see Figure 4E) by dexamethasone and combined treatment, in the absence of stimulation with IL-1α. However, we observed an upregulation of *SOCS3* upon IL-1α stimulation, as expected, but *SOCS3* levels were significantly lower in cells stimulated with IL-1α and treated with the combination, when compared to cells treated only with IL-1α (Figure 4E). In addition, we observed that STAT3-dependent mitochondrial transcription of *MT-ND1* and *MT-ND4* were significantly downregulated by the combination treatment, suggesting that dexamethasone and OLT1177 are inhibiting mitochondrial functions of STAT3 only when administered together (Figure 4F,G). To have a wider characterization of 1205Lu cells at a mitochondrial level, we decided to analyze the expression of several genes involved in different mitochondrial functions: *PPARGC1A* and *PPARGC1B* are involved in mitochondrial biogenesis, *TFAM* encodes for a mitochondrial transcription factor, *MFN1*, *MNF2* and *OPA1* regulate mitochondrial fusion, *UCP2* and *UCP3* are involved in mitochondrial homeostasis. We observed that dexamethasone downregulates the expression of *PPARGC1B* and *MFN2*, whereas the combined treatment has a negative effect on *TFAM* and *MFN2*, suggesting that the combination of dexamethasone and OLT1177 negatively affects some mitochondrial functions (Appendix A). To further characterize the effects of dexamethasone and OLT1177 on STAT3 transcriptional activity, we examined the expression levels of other STAT3 targets that rely on both STAT3 and HIF1α [73]. Of note, the transcripts *hexokinase 1* (*HK1*), *hexokinase 2* (*HK2*), and *phosphofructokinase* (*PFKP*) are each involved in glycolysis, and are significantly downregulated by the combination of dexamethasone plus OLT1177. We observed the downregulation when cells are cultured in the absence of IL-1α, but also in the presence of IL-1α (Figure 4D,E). Thus, these data highlight the marked effect that the combination of dexamethasone plus OLT1177 exerts on STAT3 transcriptional activity, compared to either single treatment alone. We conclude that the reduction in expression of these genes is part of the mechanism that accounts for the combined treatment inhibiting tumor growth, as demonstrated in Figure 1.

### 3.3. Dexamethasone and OLT1177 Negatively Affect Glycolysis in 1205Lu Cells

Malignant cells undergo metabolic changes that have been characterized as the Warburg effect. Although tumor cells are exposed to normal levels of oxygen, they preferentially use glycolysis to generate ATP. The role of inflammation and of IL-1α in the induction of aerobic glycolysis has been identified in tumors as well as in non-malignant tissues [74]. As described in Figure 5A,B, we used the SeaHorse assay to evaluate oxygen consumption rate (OCR) and glycolysis in 1205Lu cells. In 1205Lu cells treated with IL-1α, we demonstrate that the combined treatment of OLT1177 and dexamethasone negatively affects OCR in the Mito Stress Test, as well as in the ECAR of Glycolysis Stress Test (Figure 5A,B, Table 2 and Table 3). We show that IL-1α induces ATP (Figure 5C, *p* = 0.0583) as well as the glycolytic reserve (*p* < 0.05, Figure 5D). We also sought to assess the metabolic effects of the treatments upon IL-1α stimulation. Dexamethasone significantly reduces the glycolytic reserve rate, whereas OLT1177 reduces basal respiration rate (*p* = 0.0543) and ATP ratio (Figure 5E–G). The combined treatment also exerts a greater effect than either treatment alone and reduces, significantly, basal respiration, glycolytic reserve, and ATP ratio when compared to control cells (Figure 5E–G). We could not observe significant differences in glycolytic capacity, glycolysis, spare capacity, and maximal respiration between control and treated cells (Figure 5H–K). Although the effect of the treatments is clear, we could not observe a downregulation of each of the parameters measured because of the relatively short treatments. We speculate that a longer exposure of cells to dexamethasone and OLT1177 would affect the ECAR measurements more. However, we did observe that the combination of dexamethasone with OLT1177 has the opposite effect on 1205Lu cultures when these cells are cultured without IL-1α (Appendix A, Appendix A). Thus, we conclude that the combination of dexamethasone plus OLT1177 reverses the Warburg effect when tumor cells are subjected to an inflammatory signal.

## 4. Discussion

The activation of the uncontrolled STAT3 oncogene can determine the onset of various tumors [49,50,51]. Recent studies have identified a marked impact of STAT3 activity on the progression and development of melanoma, including promoting metastases [21,29,75,76,77,78,79]. Notably, most studies have focused on the nuclear activities of STAT3, with the phosphorylation of tyrosine 705. STAT3 Y705 phosphorylation is determined by Janus Kinases (JAK): JAK2 belongs to this family of kinases and the JAK2^V617F^ variant is considered as a potent oncogenic factor that leads to the constitutive activation of STAT3 and STAT5 [80,81]. The role of this JAK2 variant has mainly been studied for hematological cancers [82,83,84], but it has been identified with low frequency in melanoma as well [85]. STAT3 can additionally be phosphorylated at the level of serine 727 and this post-translational modification also affects the nuclear functions of this transcription factor [86,87,88]. However, phosphorylation of serine 727 primarily regulates the activities of the mitochondrion, inducing mitochondrial gene transcription, the functionality of the electron transport chain, and the interaction with the mitochondrial d-loop [57,89,90,91]. High levels of serine phosphorylation characterize several cancers such as gastric carcinoma [92], prostate cancer [93], epithelial carcinoma [94], breast cancer [95], and leukemia [96]. Nevertheless, few data are available on the role of serine 727 phosphorylation in melanoma progression. Serine 727 phosphorylation is induced by IFNα/γ in several human malignant melanoma lines [97] and bone marrow-derived mesenchymal stem cells, activating the LIF/ERK/pSTAT3 S727 axis, and promoting melanoma metastases [94]. Additionally, Jia et al. reported that the knockout of *Sox2* in B16F10 melanoma cells triggered the transition of cells from dormancy through the activation of either phosphorylation of tyrosine 705 or serine 727, with the subsequent activation of p53 [98].

Considering the constitutive and high levels of NLRP3 and of IL-1α in melanoma [21,99], we investigated the effects of NLRP3 inhibition on STAT3 activity and its phosphorylation. Additionally, we examined effects of NLRP3 inhibition and dexamethasone. Dexamethasone is a synthetic glucocorticoid often used to suppress inflammation in patients with autoimmune diseases [100,101], viral infections, including COVID-19 [102,103], but also in cancer patients undergoing chemotherapy [39,40]. In the present study, we combined NLRP3 inhibition and dexamethasone in mice with melanoma. We observed that NLRP3 inhibition with the orally active OLT1177 significantly reduced tumor growth of B16F10 melanoma; other studies also report the anti-tumor property of OLT1177 [21,25,29,104,105]. However, in the present study, a similar reduction in tumor growth was also observed in mice treated with dexamethasone, as shown in Figure 1. Although the demonstration of the anti-tumor properties of this glucocorticoid have also been previously reported [45,46,47,48], the combination of NLRP3 inhibition and dexamethasone resulted in a marked reduction in tumor volume compared to either treatment alone. We believe that this is the first example of combining inhibition of NLRP3 with dexamethasone.

It is likely that OLT1177 and dexamethasone target different mechanisms of STAT3 oncogenic properties. NLRP3 inhibition impacts STAT3 Y705 phosphorylation, therefore mostly affecting nuclear functions of this transcription factor and resulting in blocking of the IL-1β dependent induction of the IL-6/JAK/STAT3 axis (Appendix A) [29]. On the other hand, dexamethasone reduces STAT3 serine phosphorylation (Figure 4A), which downregulates STAT3-dependent mitochondrial transcription. We found that in the human melanoma 1205Lu cell line, as well as in primary tumors, the expression levels of the mitochondrial genes *MT-ND1* and *MT-ND4* were downregulated. The combined treatment, which negatively affects both tyrosine and serine phosphorylation of STAT3, hampers nuclear and mitochondrial functions of STAT3, resulting in inhibition of its oncogenic functions. Moreover, in the pathogenesis of melanoma, the oncogenic function of *BRAF^V600E^* mutation determines constitutive activation of the MEK/ERK pathway in 1205Lu cells, and the MEK/ERK pathway is responsible for STAT3 S727 phosphorylation [57,106,107]. We speculate that dexamethasone also negatively affects this pathway. In fact, the synergistic effects of dexamethasone with MEK/ERK inhibitors have been reported in several studies [108,109,110,111].

OLT1177 has a marked impact in STAT3-dependent transcription primarily in vivo. Nevertheless, as shown in Figure 3, the reduction in proliferation in vitro is greater with dexamethasone than OLT1177. The scratch assay demonstrated that dexamethasone suppresses cell proliferation, whereas OLT1177 does not affect this process. To show this difference, we stimulated the 1205Lu cells with IL-1α (Figure 3G). Dexamethasone also has a marked impact on STAT3-dependent gene transcription compared to OLT1177 in 1205Lu cells. However, the combined treatment, especially when cells are stimulated with IL-1α, significantly reduces STAT3 transcriptional activities. Thus, we conclude that the combination of dexamethasone plus OLT1177 is primarily due to dexamethasone inhibition of proliferation.

The effect of dexamethasone on tumor growth is likely associated with its role in preventing T cell exhaustion, a phenomenon determined by prolonged inflammation and commonly observed in cancer [112]. This is particularly relevant in the present studies, since OLT1177 reduces the level of IL-1α-induced inflammation. The reduction in tumor growth by the combination of dexamethasone plus NLRP3 blockade resulted in the optimal prevention of T cell exhaustion. Although dexamethasone reduces T cell exhaustion by inhibiting PD-L1 and idoleamine 2,3-dioxygenase [113], the dose and the duration of dexamethasone exposure in patients often have undesired effects, as recently observed in Tokunaga et al. [114], Kumar et al. [115], and Brummer et al. [116]. Our data, however, demonstrate that dexamethasone treatment, at least for the limited time of the experiment (15 days), has a beneficial effect on mice. Moreover, the additive effect of dexamethasone and OLT1177 demonstrate that this combined treatment represents an alternative strategy for melanoma treatments. The role of STAT3 in inducing PD-L1, extorting prooncogenic functions, has been demonstrated in several tumor models characterized by the JAK2^V617F^ variant. This constitutively active JAK2 form, that hyperactivates the STAT3 phosphorylation, leads to upregulation of PD-L1 and to the immune escape of neoplasms [117,118,119]. The combined treatment, by inhibiting STAT3 functions, can negatively impact the STAT3-dependent induction of PD-L1, reducing immune escape and tumor growth.

From a metabolic point of view, as depicted in Figure 5, OLT1177 blocks glycolysis to a greater extent than dexamethasone. We show that the activity of NLRP3 inhibition is enhanced by combination with dexamethasone, highlighting the additive effect of these separate mechanisms on glycolysis. Considering the data described on metabolic changes, we conclude that the effect of OLT1177 on STAT3 Y705 phosphorylation impacts primarily on glycolysis, rather than on dexamethasone-dependent inhibition of S727 phosphorylation. However, with the reduction of both Y705 and S727 phosphorylations with the combined treatment, the total effect on glycolysis is optimal.

In summary, our data show that dexamethasone enhances the anti-inflammatory property of NLRP3 inhibition by dampening the pro-oncogenic functions of STAT3. Considering the beneficial effects of inflammasome inhibition for cancer treatment, particularly for those melanoma patients that are resistant to checkpoint inhibitors [20], an alternative treatment is the combination of OLT1177 with dexamethasone. The combination enhances the anti-tumor properties of checkpoint inhibitors, but at the same time the combination lessens the inflammatory side-effects of checkpoint inhibitors during treatment.

## Figures and Tables

**Figure 1 cells-12-00294-f001:**
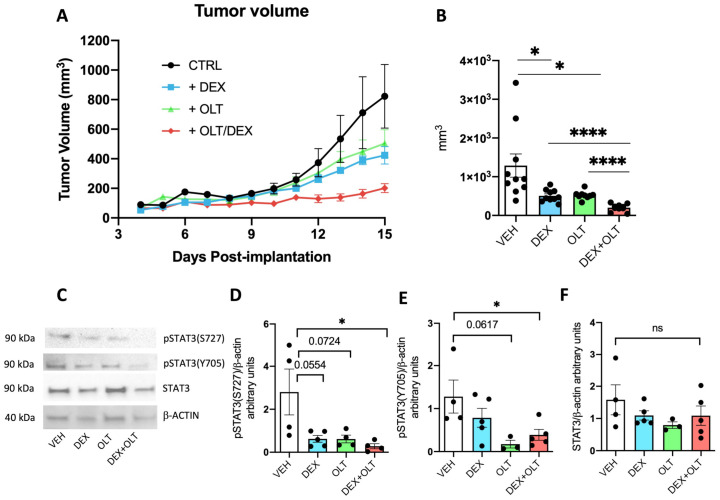
Inhibition of melanoma growth and STAT3 phosphorylations by OLT1177 and dexamethasone. (**A**) Measurement of tumor growth from day 4 post-implantation to day 15. Tumors were measured every day with an electronic caliper. (**B**) Tumor volume on day 15 derived from data in A. (**C**) A representative Western blot performed on primary tumor samples. (**D**) Histogram derived from Western blot (shown in (**C**)) of pSTAT3 (S727) levels in the primary tumor. (**E**) Histogram derived from Western blot (shown in (**C**)) of pSTAT3 (Y705) levels in the primary tumor. (**F**) Histogram derived from Western blot (shown in (**C**)) of total STAT3 levels in the primary tumor. The data are presented as mean ± SEM. * *p* < 0.05; **** *p* < 0.0001; ns = not significant.

**Figure 2 cells-12-00294-f002:**
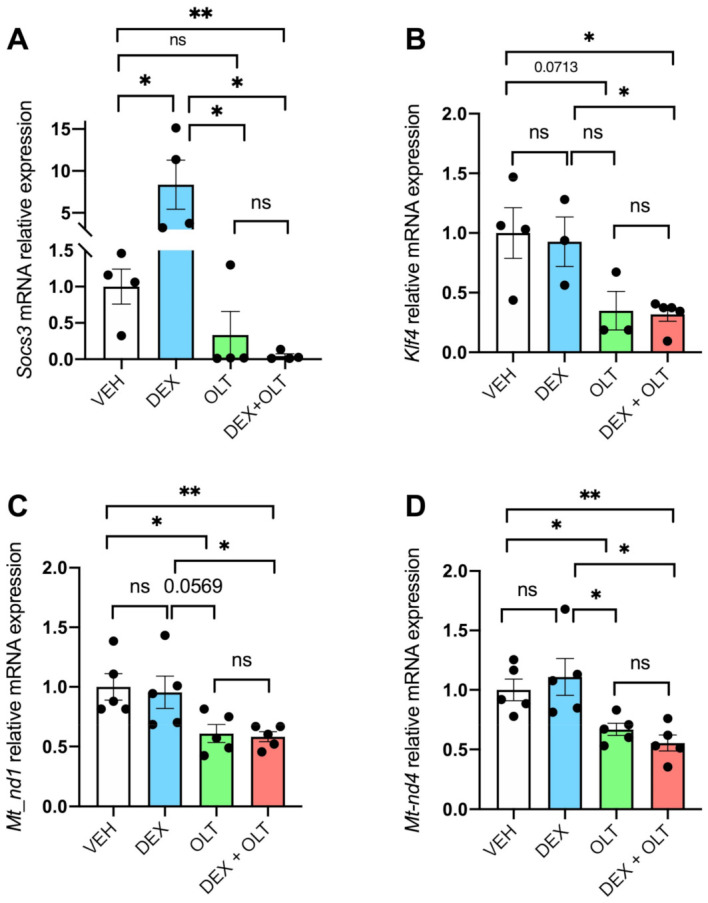
STAT3 transcriptional activity is inhibited by OLT1177 and dexamethasone. Total RNA was extracted from primary tumor samples and RT-qPCR was performed for (**A**) *Socs3*, (**B**) *Klf4*, (**C**) *Mt-nd1*, (**D**) and *Mt-nd4*. Data are presented as mean ± SEM. * *p* < 0.05; ** *p* < 0.01; ns = not significant.

**Figure 3 cells-12-00294-f003:**
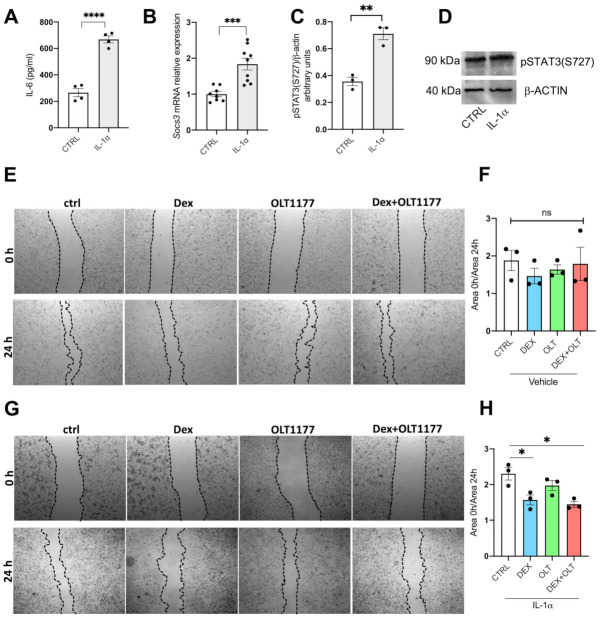
IL-1β induced proliferation is reduced by OLT1177 and dexamethasone. (**A**) IL-6 levels in the supernatants of 1205Lu incubated for 24 h with human IL-1α. (**B**) Gene expression of *SOCS3* in 1205Lu cells from (**A**). (**C**) Western blot analysis of pSTAT3 (S727) in 1205Lu cells incubated for 24 h with either vehicle of human IL-1α. (**D**) A representative Western blot performed on 1205Lu cells from cells shown in (**B**). (**E**) A representative photomicrograph of 1205Lu cells treated with vehicle, dexamethasone, OLT1177 or dexamethasone + OLT1177 at the beginning of the experiment (0 h) and after 24 h. (**F**) Quantification of the growth area measured as the ratio between the scratched area at the beginning of the experiment and at the end. (**G**) A representative photomicrograph of 1205Lu cells stimulated with human IL-1α and then treated with vehicle, dexamethasone, OLT1177 or dexamethasone plus OLT1177 at the beginning of the experiment (0 h) and at the end (24 h). (**H**) Quantification of the growth area measured as the ratio between the scratched area at the beginning of the experiment and at the end. Data are presented as mean ± SEM. * *p* < 0.05; ** *p* < 0.01; *** *p* < 0.001; **** *p* < 0.0001; ns = not significant.

**Figure 4 cells-12-00294-f004:**
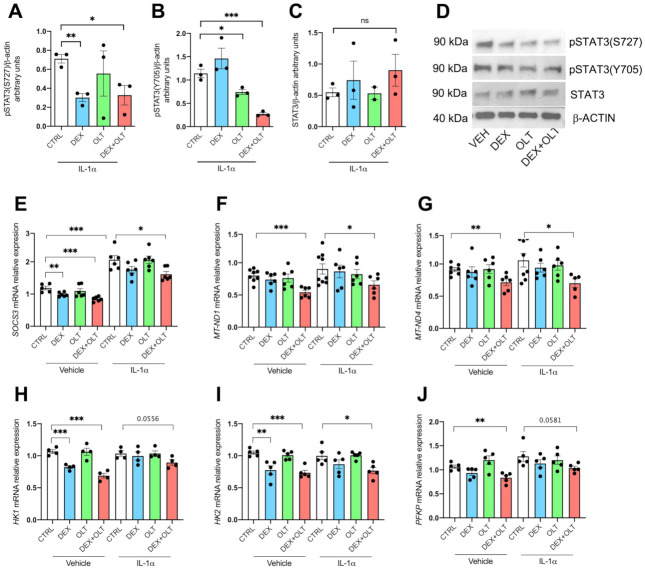
STAT3 phosphorylations and activities are inhibited by OLT1177 and dexamethasone in LU1205 cells. (**A**) Western blot of pSTAT3 (S727) levels in 1205Lu cells treated with IL-1α followed by vehicle, dexamethasone, OLT1177 or the combination of dexamethasone plus OLT1177. (**B**) A representative Western blot of pSTAT3 (Y705) levels in the same cells as in A. (**C**) A representative Western blot of total STAT3 levels cells shown in A. (**D**) Representative pictures of LU1205 lysates. The data are shown as mean ± SEM. (**E**–**I**) Gene expression of (**E**) *SOCS3*, (**F**) *MT_ND1*, (**G**) *MT_ND4*, (**H**) *HK1*, (**I**) *HK2*, and (**J**) *PFKP*, respectively, in cells stimulated with IL-1α or vehicle and then treated with IL-1α, dexamethasone, OLT1177 or dexamethasone plus OLT1177. The data are presented as mean ± SEM. * *p* < 0.05; ** *p* < 0.01; *** *p* < 0.001.

**Figure 5 cells-12-00294-f005:**
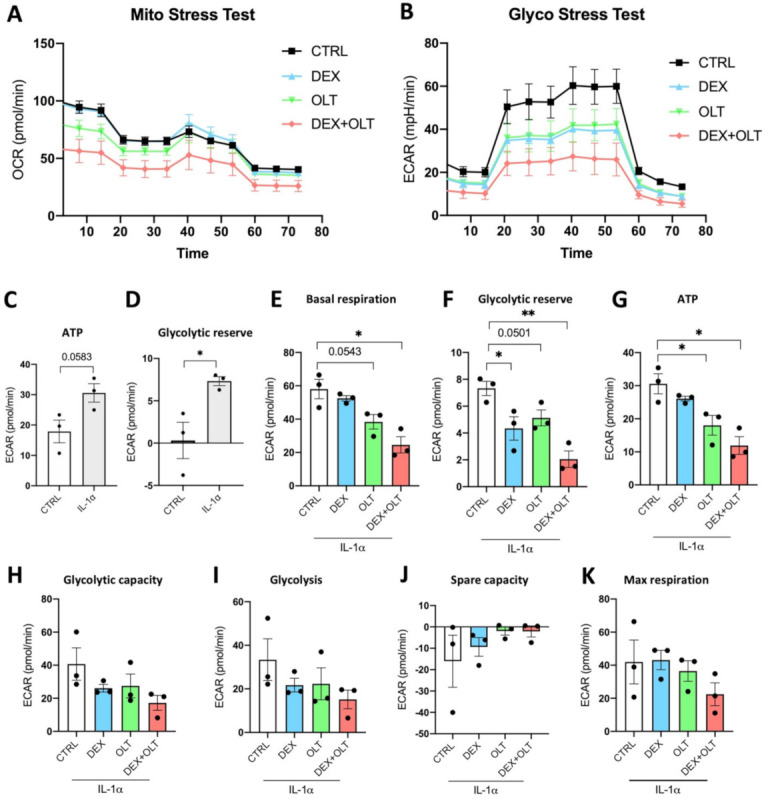
Dexamethasone and OLT1177 reduce OCR and ECAR in 1205Lu cells. (**A**) Measurement of OCR in 1205Lu stimulated with IL−1α and treated with vehicle, dexamethasone, OLT1177 or the combination of dexamethasone plus OLT1177 (Mito stress test). (**B**) Measurement of ECAR in 1205Lu treated with IL−1α plus vehicle, dexamethasone, OLT1177 and dexamethasone + OLT1177 (Glyco stress test). (**C**) ATP rate measured in control 1205Lu cells and in 1205Lu cells treated with 10 ng/mL IL−1α 16 h. (**D**) Glycolytic reserve measured in control 1205Lu cells and in 1205Lu cells treated with 10 ng/mL IL−1α for 16 h. (**E**) Basal respiration rate measured in control 1205Lu treated with 10 ng/mL IL−1α and vehicle, dexamethasone, OLT1177, dexamethasone + OLT1177 for 16 h. (**F**) Glycolytic reserve rate measured in control 1205Lu treated with 10 ng/mL IL−1α and vehicle, dexamethasone, OLT1177, dexamethasone + OLT1177 for 16 h. (**G**) ATP rate measured in control 1205Lu treated with 10 ng/mL IL−1α and vehicle, dexamethasone, OLT1177, dexamethasone + OLT1177 for 16 h. (**H**) Glycolytic capacity rate measured in control 1205Lu treated with 10 ng/mL IL−1α and vehicle, dexamethasone, OLT1177, dexamethasone + OLT1177 for 16 h. (**I**) Glycolysis rate measured in control 1205Lu treated with 10 ng/mL IL−1α and vehicle, dexamethasone, OLT1177, dexamethasone + OLT1177 for 16 h. (**J**) Spare capacity rate measured in control 1205Lu treated with 10 ng/mL IL−1α and vehicle, dexamethasone, OLT1177, dexamethasone + OLT1177 for 16 h. (**K**) Maximal respiration rate measured in control 1205Lu treated with 10 ng/mL IL−1α and vehicle, dexamethasone, OLT1177, dexamethasone + OLT1177 for 16 h. The data are presented as mean ± SEM. * *p* < 0.05; ** *p* < 0.01.

**Table 1 cells-12-00294-t001:** Statistical analysis of Figure 1A.

Tukey’s Multiple Comparisons Test	Mean Diff.	95.00% CI of Diff.	Significant?	Summary	Adjusted *p* Value
**Row 1**					
+ OLT/DEX vs. + DEX	11.65	−212.9 to 236.2	No	ns	0.9991
+ OLT/DEX vs. CTRL	−23.65	−248.2 to 200.9	No	ns	0.9929
+ OLT/DEX vs. + OLT	−1.420	−226.0 to 223.2	No	ns	>0.9999
+ DEX vs. CTRL	−35.30	−259.9 to 189.3	No	ns	0.9771
+ DEX vs. + OLT	−13.07	−237.7 to 211.5	No	ns	0.9988
CTRL vs. + OLT	22.23	−202.4 to 246.8	No	ns	0.9941
**Row 2**					
+ OLT/DEX vs. + DEX	−11.30	−235.9 to 213.3	No	ns	0.9992
+ OLT/DEX vs. CTRL	−20.68	−245.3 to 203.9	No	ns	0.9952
+ OLT/DEX vs. + OLT	−77.61	−302.2 to 147.0	No	ns	0.8072
+ DEX vs. CTRL	−9.383	−234.0 to 215.2	No	ns	0.9995
+ DEX vs. + OLT	−66.31	−290.9 to 158.3	No	ns	0.8700
CTRL vs. + OLT	−56.93	−281.5 to 167.7	No	ns	0.9130
**Row 3**					
+ OLT/DEX vs. + DEX	4.053	−220.5 to 228.6	No	ns	>0.9999
+ OLT/DEX vs. CTRL	−66.04	−290.6 to 158.5	No	ns	0.8714
+ OLT/DEX vs. + OLT	−17.50	−242.1 to 207.1	No	ns	0.9971
+ DEX vs. CTRL	−70.09	−294.7 to 154.5	No	ns	0.8502
+ DEX vs. + OLT	−21.55	−246.1 to 203.0	No	ns	0.9946
CTRL vs. + OLT	48.54	−176.0 to 273.1	No	ns	0.9437
**Row 4**					
+ OLT/DEX vs. + DEX	−18.32	−242.9 to 206.3	No	ns	0.9967
+ OLT/DEX vs. CTRL	−70.61	−295.2 to 154.0	No	ns	0.8474
+ OLT/DEX vs. + OLT	−38.04	−262.6 to 186.5	No	ns	0.9716
+ DEX vs. CTRL	−52.30	−276.9 to 172.3	No	ns	0.9309
+ DEX vs. + OLT	−19.72	−244.3 to 204.9	No	ns	0.9958
CTRL vs. + OLT	32.57	−192.0 to 257.2	No	ns	0.9818
**Row 5**					
+ OLT/DEX vs. + DEX	−42.92	−267.5 to 181.7	No	ns	0.9601
+ OLT/DEX vs. CTRL	−45.14	−269.7 to 179.4	No	ns	0.9540
+ OLT/DEX vs. + OLT	−29.61	−254.2 to 195.0	No	ns	0.9862
+ DEX vs. CTRL	−2.223	−226.8 to 222.4	No	ns	>0.9999
+ DEX vs. + OLT	13.31	−211.3 to 237.9	No	ns	0.9987
CTRL vs. + OLT	15.53	−209.1 to 240.1	No	ns	0.9979
**Row 6**					
+ OLT/DEX vs. + DEX	−43.15	−267.7 to 181.4	No	ns	0.9595
+ OLT/DEX vs. CTRL	−61.91	−286.5 to 162.7	No	ns	0.8913
+ OLT/DEX vs. + OLT	−39.84	−264.4 to 184.8	No	ns	0.9677
+ DEX vs. CTRL	−18.76	−243.3 to 205.8	No	ns	0.9964
+ DEX vs. + OLT	3.313	−221.3 to 227.9	No	ns	>0.9999
CTRL vs. + OLT	22.07	−202.5 to 246.7	No	ns	0.9942
**Row 7**					
+ OLT/DEX vs. + DEX	−85.30	−309.9 to 139.3	No	ns	0.7586
+ OLT/DEX vs. CTRL	−101.6	−326.2 to 123.0	No	ns	0.6451
+ OLT/DEX vs. + OLT	−84.29	−308.9 to 140.3	No	ns	0.7652
+ DEX vs. CTRL	−16.29	−240.9 to 208.3	No	ns	0.9976
+ DEX vs. + OLT	1.008	−223.6 to 225.6	No	ns	>0.9999
CTRL vs. + OLT	17.30	−207.3 to 241.9	No	ns	0.9972
**Row 8**					
+ OLT/DEX vs. + DEX	−62.50	−287.1 to 162.1	No	ns	0.8885
+ OLT/DEX vs. CTRL	−120.4	−345.0 to 104.2	No	ns	0.5075
+ OLT/DEX vs. + OLT	−99.77	−324.4 to 124.8	No	ns	0.6582
+ DEX vs. CTRL	−57.90	−282.5 to 166.7	No	ns	0.9090
+ DEX vs. + OLT	−37.27	−261.9 to 187.3	No	ns	0.9733
CTRL vs. + OLT	20.64	−203.9 to 245.2	No	ns	0.9952
**Row 9**					
+ OLT/DEX vs. + DEX	−136.1	−360.7 to 88.45	No	ns	0.3977
+ OLT/DEX vs. CTRL	−243.6	−468.2 to −19.05	Yes	*	0.0277
+ OLT/DEX vs. + OLT	−173.5	−398.1 to 51.06	No	ns	0.1907
+ DEX vs. CTRL	−107.5	−332.1 to 117.1	No	ns	0.6019
+ DEX vs. + OLT	−37.39	−262.0 to 187.2	No	ns	0.9730
CTRL vs. + OLT	70.11	−154.5 to 294.7	No	ns	0.8501
**Row 10**					
+ OLT/DEX vs. + DEX	−182.5	−407.1 to 42.05	No	ns	0.1547
+ OLT/DEX vs. CTRL	−396.0	−620.5 to −171.4	Yes	****	<0.0001
+ OLT/DEX vs. + OLT	−256.5	−481.1 to −31.91	Yes	*	0.0181
+ DEX vs. CTRL	−213.4	−438.0 to 11.17	No	ns	0.0691
+ DEX vs. + OLT	−73.96	−298.5 to 150.6	No	ns	0.8286
CTRL vs. + OLT	139.5	−85.13 to 364.0	No	ns	0.3758
**Row 11**					
+ OLT/DEX vs. + DEX	−226.5	−451.1 to −1.906	Yes	*	0.0472
+ OLT/DEX vs. CTRL	−548.4	−773.0 to −323.8	Yes	****	<0.0001
+ OLT/DEX vs. + OLT	−285.8	−510.4 to −61.19	Yes	**	0.0063
+ DEX vs. CTRL	−321.9	−546.5 to −97.29	Yes	**	0.0015
+ DEX vs. + OLT	−59.29	−283.9 to 165.3	No	ns	0.9030
CTRL vs. + OLT	262.6	38.00 to 487.2	Yes	*	0.0146
**Row 12**					
+ OLT/DEX vs. + DEX	−221.7	−446.3 to 2.844	No	ns	0.0544
+ OLT/DEX vs. CTRL	−621.3	−845.9 to −396.7	Yes	****	<0.0001
+ OLT/DEX vs. + OLT	−301.8	−526.4 to −77.25	Yes	**	0.0034
+ DEX vs. CTRL	−399.5	−624.1 to −174.9	Yes	****	<0.0001
+ DEX vs. + OLT	−80.09	−304.7 to 144.5	No	ns	0.7919
CTRL vs. + OLT	319.4	94.86 to 544.0	Yes	**	0.0017

* *p* < 0.05; ** *p* < 0.01; **** *p* < 0.0001; ns = not significant.

**Table 2 cells-12-00294-t002:** Statistical analysis of Figure 5A.

Tukey’s Multiple Comparisons Test	Mean Diff.	95.00% CI of Diff.	Significant?	Summary	Adjusted *p* Value
**Row 1**					
Con IL1a vs. Dex IL1a	1.240	−20.43 to 22.91	No	ns	0.9988
Con IL1a vs. OLT IL1a	19.92	1.745 to 41.59	No	ns	0.0831
Con IL1a vs. Combo IL1a	41.46	19.80 to 63.13	Yes	****	<0.0001
Dex IL1a vs. OLT IL1a	18.68	−2.985 to 40.35	No	ns	0.1162
Dex IL1a vs. Combo IL1a	40.22	18.56 to 61.89	Yes	****	<0.0001
OLT IL1a vs. Combo IL1a	21.54	−0.1282 to 43.21	No	ns	0.0520
**Row 2**					
Con IL1a vs. Dex IL1a	1.570	−20.10 to 23.24	No	ns	0.9976
Con IL1a vs. OLT IL1a	18.53	−3.142 to 40.19	No	ns	0.1211
Con IL1a vs. Combo IL1a	38.15	16.48 to 59.82	Yes	****	<0.0001
Dex IL1a vs. OLT IL1a	16.96	−4.712 to 38.62	No	ns	0.1786
Dex IL1a vs. Combo IL1a	36.58	14.91 to 58.25	Yes	***	0.0002
OLT IL1a vs. Combo IL1a	19.62	−2.045 to 41.29	No	ns	0.0903
**Row 3**					
Con IL1a vs. Dex IL1a	1.150	−20.52 to 22.82	No	ns	0.9990
Con IL1a vs. OLT IL1a	18.37	−3.298 to 40.04	No	ns	0.1261
Con IL1a vs. Combo IL1a	36.89	15.23 to 58.56	Yes	***	0.0001
Dex IL1a vs. OLT IL1a	17.22	−4.448 to 38.89	No	ns	0.1677
Dex IL1a vs. Combo IL1a	35.74	14.08 to 57.41	Yes	***	0.0002
OLT IL1a vs. Combo IL1a	18.52	−3.145 to 40.19	No	ns	0.1212
**Row 4**					
Con IL1a vs. Dex IL1a	0.7667	−20.90 to 22.43	No	ns	0.9997
Con IL1a vs. OLT IL1a	10.01	−11.66 to 31.67	No	ns	0.6237
Con IL1a vs. Combo IL1a	24.41	2.742 to 46.08	Yes	*	0.0208
Dex IL1a vs. OLT IL1a	9.240	−12.43 to 30.91	No	ns	0.6813
Dex IL1a vs. Combo IL1a	23.64	1.975 to 45.31	Yes	*	0.0268
OLT IL1a vs. Combo IL1a	14.40	−7.265 to 36.07	No	ns	0.3099
**Row 5**					
Con IL1a vs. Dex IL1a	0.2300	−21.44 to 21.90	No	ns	>0.9999
Con IL1a vs. OLT IL1a	8.733	−12.93 to 30.40	No	ns	0.7182
Con IL1a vs. Combo IL1a	24.18	2.515 to 45.85	Yes	*	0.0224
Dex IL1a vs. OLT IL1a	8.503	−13.16 to 30.17	No	ns	0.7346
Dex IL1a vs. Combo IL1a	23.95	2.285 to 45.62	Yes	*	0.0242
OLT IL1a vs. Combo IL1a	15.45	−6.218 to 37.12	No	ns	0.2503
**Row 6**					
Con IL1a vs. Dex IL1a	0.5400	−21.13 to 22.21	No	ns	>0.9999
Con IL1a vs. OLT IL1a	9.023	−12.64 to 30.69	No	ns	0.6972
Con IL1a vs. Combo IL1a	24.24	2.575 to 45.91	Yes	*	0.0220
Dex IL1a vs. OLT IL1a	8.483	−13.18 to 30.15	No	ns	0.7360
Dex IL1a vs. Combo IL1a	23.70	2.035 to 45.37	Yes	*	0.0263
OLT IL1a vs. Combo IL1a	15.22	−6.448 to 36.89	No	ns	0.2628
**Row 7**					
Con IL1a vs. Dex IL1a	−7.437	−29.10 to 14.23	No	ns	0.8062
Con IL1a vs. OLT IL1a	1.087	−20.58 to 22.75	No	ns	0.9992
Con IL1a vs. Combo IL1a	20.30	−1.372 to 41.96	No	ns	0.0748
Dex IL1a vs. OLT IL1a	8.523	−13.14 to 30.19	No	ns	0.7332
Dex IL1a vs. Combo IL1a	27.73	6.065 to 49.40	Yes	**	0.0063
OLT IL1a vs. Combo IL1a	19.21	−2.458 to 40.88	No	ns	0.1011
**Row 8**					
Con IL1a vs. Dex IL1a	−5.763	−27.43 to 15.90	No	ns	0.8986
Con IL1a vs. OLT IL1a	−0.5900	−22.26 to 21.08	No	ns	0.9999
Con IL1a vs. Combo IL1a	16.78	−4.885 to 38.45	No	ns	0.1860
Dex IL1a vs. OLT IL1a	5.173	−16.49 to 26.84	No	ns	0.9241
Dex IL1a vs. Combo IL1a	22.55	0.8784 to 44.21	Yes	*	0.0381
OLT IL1a vs. Combo IL1a	17.37	−4.295 to 39.04	No	ns	0.1617
**Row 9**					
Con IL1a vs. Dex IL1a	−3.410	−25.08 to 18.26	No	ns	0.9764
Con IL1a vs. OLT IL1a	0.6200	−21.05 to 22.29	No	ns	0.9998
Con IL1a vs. Combo IL1a	16.63	−5.035 to 38.30	No	ns	0.1926
Dex IL1a vs. OLT IL1a	4.030	−17.64 to 25.70	No	ns	0.9620
Dex IL1a vs. Combo IL1a	20.04	−1.625 to 41.71	No	ns	0.0804
OLT IL1a vs. Combo IL1a	16.01	−5.655 to 37.68	No	ns	0.2216
**Row 10**					
Con IL1a vs. Dex IL1a	3.163	−18.50 to 24.83	No	ns	0.9810
Con IL1a vs. OLT IL1a	5.193	−16.47 to 26.86	No	ns	0.9233
Con IL1a vs. Combo IL1a	14.96	−6.712 to 36.62	No	ns	0.2775
Dex IL1a vs. OLT IL1a	2.030	−19.64 to 23.70	No	ns	0.9948
Dex IL1a vs. Combo IL1a	11.79	−9.875 to 33.46	No	ns	0.4882
OLT IL1a vs. Combo IL1a	9.763	−11.90 to 31.43	No	ns	0.6421
**Row 11**					
Con IL1a vs. Dex IL1a	2.587	−19.08 to 24.25	No	ns	0.9894
Con IL1a vs. OLT IL1a	5.037	−16.63 to 26.70	No	ns	0.9294
Con IL1a vs. Combo IL1a	14.58	−7.085 to 36.25	No	ns	0.2991
Dex IL1a vs. OLT IL1a	2.450	−19.22 to 24.12	No	ns	0.9910
Dex IL1a vs. Combo IL1a	12.00	−9.672 to 33.66	No	ns	0.4732
OLT IL1a vs. Combo IL1a	9.547	−12.12 to 31.21	No	ns	0.6584
**Row 12**					
Con IL1a vs. Dex IL1a	2.880	−18.79 to 24.55	No	ns	0.9855
Con IL1a vs. OLT IL1a	4.993	−16.67 to 26.66	No	ns	0.9310
Con IL1a vs. Combo IL1a	14.38	−7.292 to 36.04	No	ns	0.3116
Dex IL1a vs. OLT IL1a	2.113	−19.55 to 23.78	No	ns	0.9941
Dex IL1a vs. Combo IL1a	11.50	−10.17 to 33.16	No	ns	0.5104
OLT IL1a vs. Combo IL1a	9.383	−12.28 to 31.05	No	ns	0.6706

* *p* < 0.05; ** *p* < 0.01; *** *p* < 0.001; **** *p* < 0.0001; ns = not significant.

**Table 3 cells-12-00294-t003:** Statistical analysis of Figure 5B.

Tukey’s Multiple Comparisons Test	Mean Diff.	95.00% CI of Diff.	Significant?	Summary	Adjusted *p* Value
**Row 1**					
Con IL1a vs. Dex IL1a	7.183	−10.32 to 24.68	No	ns	0.7066
Con IL1a vs. OLT IL1a	6.860	−10.64 to 24.36	No	ns	0.7353
Con IL1a vs. Combo IL1a	13.09	−4.410 to 30.59	No	ns	0.2121
Dex IL1a vs. OLT IL1a	−0.3233	−17.82 to 17.18	No	ns	>0.9999
Dex IL1a vs. Combo IL1a	5.907	−11.59 to 23.41	No	ns	0.8139
OLT IL1a vs. Combo IL1a	6.230	−11.27 to 23.73	No	ns	0.7884
**Row 2**					
Con IL1a vs. Dex IL1a	5.567	−11.93 to 23.07	No	ns	0.8393
Con IL1a vs. OLT IL1a	4.950	−12.55 to 22.45	No	ns	0.8809
Con IL1a vs. Combo IL1a	9.573	−7.926 to 27.07	No	ns	0.4837
Dex IL1a vs. OLT IL1a	−0.6167	−18.12 to 16.88	No	ns	0.9997
Dex IL1a vs. Combo IL1a	4.007	−13.49 to 21.51	No	ns	0.9322
OLT IL1a vs. Combo IL1a	4.623	−12.88 to 22.12	No	ns	0.9004
**Row 3**					
Con IL1a vs. Dex IL1a	5.733	−11.77 to 23.23	No	ns	0.8270
Con IL1a vs. OLT IL1a	5.070	−12.43 to 22.57	No	ns	0.8732
Con IL1a vs. Combo IL1a	9.850	−7.650 to 27.35	No	ns	0.4585
Dex IL1a vs. OLT IL1a	−0.6633	−18.16 to 16.84	No	ns	0.9996
Dex IL1a vs. Combo IL1a	4.117	−13.38 to 21.62	No	ns	0.9271
OLT IL1a vs. Combo IL1a	4.780	−12.72 to 22.28	No	ns	0.8912
**Row 4**					
Con IL1a vs. Dex IL1a	15.60	−1.896 to 33.10	No	ns	0.0981
Con IL1a vs. OLT IL1a	14.36	−3.140 to 31.86	No	ns	0.1463
Con IL1a vs. Combo IL1a	26.28	8.780 to 43.78	Yes	***	0.0009
Dex IL1a vs. OLT IL1a	−1.243	−18.74 to 16.26	No	ns	0.9977
Dex IL1a vs. Combo IL1a	10.68	−6.823 to 28.18	No	ns	0.3862
OLT IL1a vs. Combo IL1a	11.92	−5.580 to 29.42	No	ns	0.2888
**Row 5**					
Con IL1a vs. Dex IL1a	17.24	−0.2629 to 34.74	No	ns	0.0551
Con IL1a vs. OLT IL1a	15.72	−1.783 to 33.22	No	ns	0.0944
Con IL1a vs. Combo IL1a	28.10	10.60 to 45.60	Yes	***	0.0003
Dex IL1a vs. OLT IL1a	−1.520	−19.02 to 15.98	No	ns	0.9958
Dex IL1a vs. Combo IL1a	10.86	−6.640 to 28.36	No	ns	0.3709
OLT IL1a vs. Combo IL1a	12.38	−5.120 to 29.88	No	ns	0.2568
**Row 6**					
Con IL1a vs. Dex IL1a	17.42	−0.07626 to 34.92	No	ns	0.0514
Con IL1a vs. OLT IL1a	15.80	−1.696 to 33.30	No	ns	0.0917
Con IL1a vs. Combo IL1a	27.47	9.967 to 44.97	Yes	***	0.0005
Dex IL1a vs. OLT IL1a	−1.620	−19.12 to 15.88	No	ns	0.9950
Dex IL1a vs. Combo IL1a	10.04	−7.456 to 27.54	No	ns	0.4412
OLT IL1a vs. Combo IL1a	11.66	−5.836 to 29.16	No	ns	0.3076
**Row 7**					
Con IL1a vs. Dex IL1a	20.19	2.690 to 37.69	Yes	*	0.0170
Con IL1a vs. OLT IL1a	18.45	0.9504 to 35.95	Yes	*	0.0347
Con IL1a vs. Combo IL1a	32.89	15.39 to 50.39	Yes	****	<0.0001
Dex IL1a vs. OLT IL1a	−1.740	−19.24 to 15.76	No	ns	0.9938
Dex IL1a vs. Combo IL1a	12.70	−4.800 to 30.20	No	ns	0.2360
OLT IL1a vs. Combo IL1a	14.44	−3.060 to 31.94	No	ns	0.1427
**Row 8**					
Con IL1a vs. Dex IL1a	20.47	2.967 to 37.97	Yes	*	0.0151
Con IL1a vs. OLT IL1a	17.85	0.3537 to 35.35	Yes	*	0.0437
Con IL1a vs. Combo IL1a	33.42	15.92 to 50.92	Yes	****	<0.0001
Dex IL1a vs. OLT IL1a	−2.613	−20.11 to 14.89	No	ns	0.9797
Dex IL1a vs. Combo IL1a	12.95	−4.546 to 30.45	No	ns	0.2203
OLT IL1a vs. Combo IL1a	15.57	−1.933 to 33.07	No	ns	0.0993
**Row 9**					
Con IL1a vs. Dex IL1a	20.41	2.914 to 37.91	Yes	*	0.0154
Con IL1a vs. OLT IL1a	17.61	0.1071 to 35.11	Yes	*	0.0480
Con IL1a vs. Combo IL1a	34.00	16.50 to 51.50	Yes	****	<0.0001
Dex IL1a vs. OLT IL1a	−2.807	−20.31 to 14.69	No	ns	0.9750
Dex IL1a vs. Combo IL1a	13.59	−3.910 to 31.09	No	ns	0.1841
OLT IL1a vs. Combo IL1a	16.40	−1.103 to 33.90	No	ns	0.0747
**Row 10**					
Con IL1a vs. Dex IL1a	6.900	−10.60 to 24.40	No	ns	0.7318
Con IL1a vs. OLT IL1a	5.840	−11.66 to 23.34	No	ns	0.8190
Con IL1a vs. Combo IL1a	11.05	−6.453 to 28.55	No	ns	0.3556
Dex IL1a vs. OLT IL1a	−1.060	−18.56 to 16.44	No	ns	0.9986
Dex IL1a vs. Combo IL1a	4.147	−13.35 to 21.65	No	ns	0.9256
OLT IL1a vs. Combo IL1a	5.207	−12.29 to 22.71	No	ns	0.8643
**Row 11**					
Con IL1a vs. Dex IL1a	5.283	−12.22 to 22.78	No	ns	0.8591
Con IL1a vs. OLT IL1a	5.103	−12.40 to 22.60	No	ns	0.8711
Con IL1a vs. Combo IL1a	9.153	−8.346 to 26.65	No	ns	0.5227
Dex IL1a vs. OLT IL1a	−0.1800	−17.68 to 17.32	No	ns	>0.9999
Dex IL1a vs. Combo IL1a	3.870	−13.63 to 21.37	No	ns	0.9384
OLT IL1a vs. Combo IL1a	4.050	−13.45 to 21.55	No	ns	0.9302
**Row 12**					
Con IL1a vs. Dex IL1a	4.440	−13.06 to 21.94	No	ns	0.9106
Con IL1a vs. OLT IL1a	4.453	−13.05 to 21.95	No	ns	0.9098
Con IL1a vs. Combo IL1a	7.860	−9.640 to 25.36	No	ns	0.6444
Dex IL1a vs. OLT IL1a	0.01333	−17.49 to 17.51	No	ns	>0.9999
Dex IL1a vs. Combo IL1a	3.420	−14.08 to 20.92	No	ns	0.9563
OLT IL1a vs. Combo IL1a	3.407	−14.09 to 20.91	No	ns	0.9568

* *p* < 0.05; *** *p* < 0.001; **** *p* < 0.0001; ns = not significant.

## Data Availability

Not applicable.

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
