# Peer review of "Dexamethasone and OLT1177 Cooperate in the Reduction of Melanoma Growth by Inhibiting STAT3 Functions"

_cells, 2023, doi:10.3390/cells12020294_

Round 1

Reviewer 1 Report (Previous Reviewer 1)

The authors have addressed all my comments. I would like to endorse the manuscript for the publication.

Author Response

We want to thank the reviewer for its positive comment and for endorsing the manuscript for publication. 

Reviewer 2 Report (Previous Reviewer 2)

Comments 2:

I am satisfied with the revision performed by the reviewers. However, I have some minor submissions/suggestions.

1.     Have authors studied the antitumor effect of OLT1177 + anti-PD-1 vs their combination OLT1177 + Dexamethasone. I am just curious whether there is any additive anti-tumor effects imparted by the combination that authors have used in this study vs the OLT1177 + anti-PD-1 combination.

2.     Some sentences like in page 2, paragraph 2 and line 57 “in humans, human administration of oral…….” seems like “in humans oral administration….” There are more grammatical corrections in the text which authors should go through and correct them. Though, the manuscript is written in good language.

I think the manuscript should be good to go for publication after these minor corrects.

Thanks

Author Response

We thank the reviewer for the positive comments. 

  1. We thank the reviewer for raising such an interesting question. We compared the effects observed by Tengesdal and collaborators (2021, PNAS) with the results obtained in this study. The OLT1177+aPD-1 combination determines a 70% reduction of tumor growth when compared to vehicle, whereas OLT1177+dexamethasone results in a 75% reduction compared to vehicle. This rough comparison suggests that in a 15-day long trial, OLT1177 is more effective if combined with dexamethasone rather than with aPD-1. However, to give a more accurate answer to this question we should perform the two treatments in the same experiment and we think that it is far beyond our aim. These two treatments have been though for two different rationals: OLT1177+aPD-1 was used to dampen the side effects of checkpoint inhibitors, whereas with OLT1177+dexamethasone we wanted to see possible additive effects, giving that dexamethasone is usually used as an anti-inflammatory compound, especially for cancer patients. 
  2. We went through the whole manuscript and me made some changes, as highlighted by the reviewer. 

This manuscript is a resubmission of an earlier submission. The following is a list of the peer review reports and author responses from that submission.

Round 1

Reviewer 1 Report

The manuscript by Dinarello et al demonstrated entitled “Dexamethasone and OLT1177 cooperate in the reduction of melanoma growth by inhibiting STAT3 functions”. The authors first observed reduced melanoma growth with combination of OLT & DEX when compared with control. This combination inactivates the STAT3 function and reduces pSTAT3 (S727 and Y705) levels. Which concludes and correlates with reduction in the tumor growth is STAT3 dependent. The combination enhances the anti-tumor properties of checkpoint inhibitors but at the same time the combination lessens the inflammatory side-effects of checkpoint inhibitors during treatment. However, authors should address below comments before publication.

Major comments:

1.       HIF1α is a target gene for STAT3, and which regulates tumor growth. Did authors check the expression levels of HIF1.

2.       Table 1 can be moved to supplementary information

3.       The statistical analysis and significance for Fig 3F should be presented.

4.       In Fig5, Statistical analysis with significance should be presented.

5.       Does DEX, OLT, DEX+ OLT did change any morphology of mitochondria and its membrane potential?

6.       Does vessel integrity in the tumors change with drug combinations?

7.       Does authors observe any cell death or apoptosis with respect to the OLT & Dex combination?

Minor comments

Authors should check for English writing.
